# Ultrasound-guided fine needle aspiration cytology and ultrasound examination of thyroid nodules in the UAE: A comparison

Suhail Al-Salam[1], Charu Sharma[2], Maysam T. Abu Sa'a[3], Bachar Afandi[4], Khaled M. Aldahmani[4], Alia Al Dhaheri[5], Hayat Yahya[5], Duha Al Naqbi[5], Esraa Al Zuraiqi[5], Baraa Kamal Mohamed[5], Shamsa Ahmed Almansoori[5], Meera Al Zaabi[5], Aysha Al Derei[5], Amal Al Shamsi[2], Juma Al Kaabi [2]*

1 Department of Pathology, College of Medicine& Health Sciences, United Arab Emirates University, Al Ain, United Arab Emirates, 2 Department of Internal Medicine, College of Medicine& Health Sciences, United Arab Emirates University, Al Ain, United Arab Emirates, 3 Radiology Department–Tawam Hospital, Al Ain, United Arab Emirates, 4 Endocrine Division–Tawam Hospital, Al Ain, United Arab Emirates, 5 College of Medicine& Health Sciences, United Arab Emirates University, Al Ain, United Arab Emirates

* j.kaabi@uaeu.ac.ae

## Abstract

### Background

Thyroid nodules are a common clinical finding and most are benign, however, 5–15% can be malignant. There is limited regional data describing the accuracy of ultrasound-guided fine needle aspiration (FNA) cytological examination compared to ultrasound examination of thyroid in patients who have undergone thyroid surgery.

### Methods

A retrospective analysis of ultrasonographic (US) reports, FNA cytology reports and histo-pathology reports of 161 thyroid nodules presented at the endocrine center at Tawam hospital in Al Ain city, the United Arab Emirates during the period 2011–2019 was performed. US reports and images with FNA cytopathology reports and slides were reviewed by an independent radiologist and pathologist.

### Results

In total, 40 nodules were reported as benign by US examination, while very low suspicious, low suspicious, intermediate suspicious and highly suspicious categories were reported in 21, 41, 14 and 45 nodules respectively. In addition, 68 nodules were reported as benign (Bethesda category II), while atypical follicular cells of unknown significance (Bethesda category III), follicular neoplasm (Bethesda category IV), suspicious for malignancy (Bethesda category V), and malignant (Bethesda category VI) categories were reported in 33, 9, 24 and 27 nodules respectively. The risk of malignancy for US benign nodules was 5%, while the risks of malignancy in very low suspicious, low suspicious, intermediate suspicious and highly suspicious nodules were 52%, 36%, 100% and 87%, respectively. The risk of malignancy for Bethesda category II was 3%, while the risks of malignancy in category III, IV, V and VI were 58%, 67%, 96% and 100%, respectively.

**Data Availability Statement:** All relevant data are within the paper. Further demographic or detailed patient characteristics can be available upon a

reasonable request. Al Ain Medical District Human Research Ethics Committee (AAMDHREC) has imposed these restrictions. Requests for the data can be sent to Dr Rami Beiram: Assistant Dean for Research and Graduate Studies College of Medicine and Health Sciences P.O.Box: 15551, Al Ain, UAE Email: rbeiram@uaeu.ac.ae Tel: +971 3 713 7174.

**Funding:** This work was supported by research grant (SURE Grant 31M352 and 31 M355), from Research Office, UAE University. The funders had no role in study design, data collection and analysis, decision to publish, or preparation of the manuscript. The authors received no specific funding for this work.

**Competing interests:** The authors have declared that no competing interests exist.

## Conclusion

Thyroid FNA cytological examination and ultrasonography are key tools in predicting malignancy in thyroid nodules. Thyroid nodules with the diagnosis of Bethesda category III & IV run a high risk of malignancy thus more vigilance is required.

## Introduction

Nodules in thyroid glands are a common clinical finding [1]. Most thyroid nodules are benign, however, 5–15% may be malignant [1]. Thyroid nodules are more commonly seen in females and their incidence is increased with age and iodine deficiency. The use of ultrasound has improved the rate of detection and has led to a higher incidence rate. A significant number of thyroid nodules have been detected at autopsy [2]. Since the vast majority of thyroid nodules are asymptomatic, regular clinical and ultrasound (US) follow-up are essential to determine the need for further fine needle aspiration (FNA) cytology study [3]. US stratification tends to lead the treating physician to request FNA of the thyroid gland.

The Bethesda system for reporting thyroid cytopathology (TBSRTC) is the recommended methodology in reporting FNA cytology [4, 5]. TBSRTC consists of six categories and each category carries a certain risk of malignancy.

Till now, clinical management of the Bethesda category III atypia of undetermined significance (AUS) or Follicular lesion of undetermined significance (FLUS) and category IV follicular neoplasm/suspicion for a follicular neoplasm pose a great challenge between observation and follow-up with FNA with or without molecular markers, based on availability, or surgery.

Thyroid US is an important tool for the assessment of thyroid nodules. It can determine the size, site, shape, consistency, contour, circumscription, and extension to adjacent thyroid parenchyma. The American Thyroid Association (ATA) guidelines for ultrasound examination has introduced five US patterns of thyroid nodules. The benign, very low suspicion, low suspicion, intermediate suspicion, and high suspicion patterns [3]. Each category has a certain risk of malignancy [3].

There is limited regional data of thyroid nodules describing the accuracy of US thyroid in comparison to FNA of the thyroid in patients who underwent thyroid surgery. In this study, we will determine the risk of malignancy of each category in TBSRTC and ATA guidelines for thyroid sonography of thyroid nodules presented to the endocrine division at a tertiary care hospital. We will also compare the accuracy of predicting malignancy between US ATA highly suspicious reports and Bethesda category VI (malignant) reports.

## Materials and methods

A retrospective analysis was performed on patients with thyroid nodules at Tawam Hospital, Al Ain, UAE from 2011 to 2019. All patients underwent US examination, US-guided FNA cytology examinations and followed by thyroidectomy. A total number of 161 cases were found with mean age ± SE of 39.95 ± 11.49 years.

### Ethical approval

The protocol of the present study conformed to the ethical guidelines of the World Medical Association, Declaration of Helsinki, and was approved by Al Ain Medical District Human Research Ethics Committee (THREC-438). Patients or their caregivers signed a written consent allowing the use of their anonymized material for research purposes.

## Data collection

A chart review was performed and data were collected from medical records at Tawam Hospital, for the period 2011 to 2019. All adult patients (i.e., aged ≥18 years) with thyroid nodules who underwent FNA cytology and US examination for single or multiple thyroid nodules and were subjected to thyroid surgery were included in this study. The collected data including demographics, FNA reports, histopathology reports, and US reports were collected from the patients' files in the electronic medical record system.

The US thyroid images were reviewed by an independent radiologist. The FNA thyroid and the histopathology specimen were reviewed by an independent pathologist.

## Inclusion criteria

1. Adult patients with thyroid nodule detected clinically and confirmed by US examination and who underwent FNA cytology examination followed by thyroidectomy.

2. Adult patients with incidental thyroid nodules detected and confirmed by US examination and who underwent FNA cytology examination followed by thyroidectomy.

## Exclusion criteria

1. Adult patients who have thyroid nodules and underwent US and FNA cytology examination but did not have thyroidectomy with histopathologic diagnosis.

2. Adult patients with incomplete data were not included in this study.

3. Adult patients with non-diagnostic category in the Bethesda scoring system were also excluded from this study because they were not followed by thyroidectomy.

4. Adult patients with previous thyroid surgery prior to FNA or US.

5. Adult patients who have FNA or US reports of more than one year from the date of surgery.

6. Adult patients with thyroidectomy that have non-thyroid follicular cells associated pathology.

## Ultrasound assessment and categories

Ultrasound reports and images of 161 patients with thyroid nodules were retrieved from Tawam Hospital patients' files. Thyroid nodules were classified according to American thyroid association (ATA) guidelines of US assessment of thyroid nodules [3]. The US results were then compared with histopathology results.

Thyroid nodules were categorized according to the following:

- Benign: Purely cystic nodules (no solid component).

- Very low suspicion: Spongiform or partially cystic nodules without any of the US features described in low, intermediate, or high suspicion patterns.

- Low suspicion: Isoechoic or hyperechoic solid nodule, or partially cystic nodule with eccentric solid areas.

- Intermediates suspicion: Hypoechoic solid nodule with smooth margins.

- Highly suspicious: Solid hypoechoic nodule or Solid hypoechoic component of a partially cystic nodule with one or more of the following features: irregular margins (infiltrative, microlobulated), microcalcifications, taller than wide shape, rim calcifications with small extrusive soft tissue component, evidence of extra-thyroid extension

## Fine needle aspiration cytology assessment and categories

Reports and slides of 161 patients with thyroid nodules were retrieved from Tawam Hospital patients' files. FNA adequacy is defined as the presence of at least five groups of follicular cells each with 12 cells. The FNA results were then compared with histopathology results.

Thyroid nodules were classified according to The Bethesda reporting system for thyroid cytopathology and were scored according to the following [3]:

I. Non-Diagnostic or unsatisfactory

II. Benign: A non-neoplastic FNA CYTOLOGY includes colloid nodules, chronic autoimmune thyroiditis and adenomatoid nodules.

III. Atypia of undetermined significance (AUS) or Follicular lesion of undetermined

    a. significance (FLUS)

IV. Follicular neoplasm/suspicious for follicular neoplasm

V. Suspicious for malignancy

VI. Malignant

The non-diagnostic or unsatisfactory (category I) was excluded from this study because there were no thyroid surgery or histopathological examination.

## Histological classification

The histologic diagnosis of nodules was classified into 3 categories:

1. Non-neoplastic conditions: Adenomatous nodules, colloid nodules, lymphocytic thyroiditis

2. Follicular adenoma

3. Carcinoma

## Data analysis

The data was extracted from the FNA cytology, USS and histopathology reports of the patients who have undergone surgery during the study period. All data was analyzed using the Statistical Package for the Social Sciences (version 20.0) using descriptive statistics. The chi-square test with Yates correction was used to compare between frequencies. Data are presented as mean ± standard error of the mean. P values < 0.05 were considered significant.

The accuracy test was done using the following formula:

$$\text{Accuracy} = \text{True positive} + \text{True negative}/\text{True positive} + \text{True negative} + \text{False positive} + \text{False negative}.$$

Histological diagnosis was taken as the gold standard and the FNA cytology and US diagnoses were compared to it.

**Table 1. Ultrasound categories of 161 thyroid nodules according to American thyroid association guidelines.**

| Ultrasound Category | Number | Percent |
|---|---|---|
| Benign | 40 | 24.8 |
| Very low suspicion | 21 | 13.0 |
| Low suspicion | 41 | 25.6 |
| Intermediate suspicion | 14 | 8.7 |
| Highly suspicion | 45 | 27.9 |
| Total | 161 | 100 |

## Results

In total, 161 cases of thyroid nodules were studied. The mean age was 39.95 ± 11.49. Females (137) constitute 85% while males (24) constitute 15% of the cases. The mean BMI was 29.19 ± 6.32.

### Thyroid ultrasound study

In total, 40 (24.8%) nodules were reported as benign, while very low suspicious, low suspicious, intermediate suspicious, and highly suspicious comprised 21 (13%), 41 (25.6%), 14 (8.7%) and 45 (27.9%), respectively Table 1.

### Thyroid fine needle aspiration cytology

In total, 68 (42.2%) nodules were reported as benign (Bethesda category II), while AUS/FLUS (Bethesda category III), Follicular neoplasm/suspicious for follicular neoplasm
(Bethesda category IV), suspicious for malignancy (Bethesda category V), and malignant (Bethesda category VI) comprised 33 (20.5%), 9 (5.5%), 24 (15%) and 27 (16.8%), respectively Table 2.

### Histopathologic diagnosis of thyroid nodules

In total, 73 (38.4%) nodules were reported as non-neoplastic nodules (Adenomatous nodules, colloid nodules, lymphocytic thyroiditis), while follicular adenomas and carcinomas comprised 10 (6.1%) and 78 (55.5%), respectively Table 3.

### Correlation between Ultrasonographic category and risk of malignancy

The risk of malignancy for US benign category was 5%, while very low suspicious, low suspicious, intermediate suspicious, and highly suspicious were 52%, 36%, 100% and 87%, respectively, Table 4.

**Table 2. The Bethesda reporting system for thyroid cytopathology categories of 161 thyroid nodules.**

| Bethesda Category | Number | Percent |
|---|---|---|
| Benign | 68 | 42.2 |
| AUS/FLUS | 33 | 20.5 |
| Follicular neoplasm/suspicious for follicular neoplasm | 9 | 5.5 |
| Suspicious for malignancy | 24 | 15 |
| Malignant | 27 | 16.8 |
| Total | 161 | 100 |

**Table 3. Histopathologic diagnosis of 161 thyroid nodules.**

| Histologic diagnosis | Number | Percent |
|---|---|---|
| Non-neoplastic nodules | 73 | 45.3 |
| Follicular adenoma | 10 | 6.3 |
| Thyroid carcinoma | 78 | 48.4 |
| Total | 161 | 100% |

## Correlation between Bethesda categories and risk of malignancy

The risk of malignancy for Bethesda category II was 3%, while the risk of malignancy in category III, IV, V and VI were 58%, 67%, 96% and 100%, respectively Table 5.

## Comparison between Ultrasound and FNA cytology risk of malignancy

There was no significant difference between the benign category in the TBSRTC and ATA US reporting systems. The chi-square statistic was 0.4631. The *p*-value was 0.4. The chi-square statistic with Yates correction was 0.1. The *p*-value is 0.75.

There was a significant difference in the risk of malignancy between the Bethesda category VI (malignant) and the Ultrasound highly suspicious category. The chi-square statistic was 13.9. The *p*-value was 0.0002. The chi-square statistic with Yates correction was 11.85. The *p*-value was 0.0006.

## Correlation between Bethesda Categories III and IV and histologic types of carcinoma

Eighteen thyroid nodules with a cytopathologic diagnosis of category III had a histologic diagnosis of thyroid carcinoma. Interestingly, 41% (7) of them were microcarcinoma, 22% (4) were follicular variant PTC, 11% (2) were follicular carcinoma and 28% (5) were classic PTC with a tumor size pT1b (>1cm<2cm) Table 6.

Moreover, 6 thyroid nodules with a cytopathologic diagnosis of Category IV, have a histologic diagnosis of thyroid carcinoma. Interestingly, 50% (3) of them were microcarcinoma, 34% (2) were follicular variant PTC, and 16% (1) were classic PTC with a tumor size pT1b (>1cm<2cm) Table 6.

## Comparison between accuracy of predicting malignancy between Bethesda category VI and US ATA highly suspicious category

Accuracy of FNA reports Bethesda category VI = 27+0/ 27+0+0+0 = 1 = 100%

Accuracy of US ATA reports highly suspicious category = 39+0/ 39+0+6+0 = 0.87 = 87%

FNA cytological reporting of Bethesda category VI is significantly more accurate than US ATA reporting highly suspicious category, Chi-square with Yates' correction 11.85, p = 0.0006.

**Table 4. Correlation between ultrasonographic category and risk of malignancy.**

| US category | Non-Neoplastic Nodule % | Follicular adenoma % | Carcinoma % |
|---|---|---|---|
| Benign | 90 | 5 | 5 |
| Very low suspicion | 43 | 5 | 52 |
| Low suspicion | 54 | 10 | 36 |
| Intermediate suspicion | 0 | 0 | 100 |
| Highly suspicious | 11 | 2 | 87 |

**Table 5. Correlation between Bethesda categories and risk of malignancy.**

| Bethesda category | Non-Neoplastic Nodule | Follicular adenoma | Carcinoma |
|---|---|---|---|
| | % | % | % |
| II | 97 | 0 | 3 |
| III | 24 | 18 | 58 |
| IV | 22 | 11 | 67 |
| V | 0 | 4 | 96 |
| VI | 0 | 0 | 100 |

## Discussion

Many studies have considered FNA cytology as the gold standard in analyzing thyroid nodules as well as outlining the future management of patients with thyroid nodules [6–12]. However, indeterminate cytopathological diagnoses, Bethesda category III and IV, creates a problem in determining the appropriate method of management of patients with thyroid nodules.

We have shown 68 (42.2%) of FNA cytology have been reported as benign (Bethesda category II). Of these 2 were found to be malignant on histologic examination, giving a 3% risk of malignancy. This is mainly related to miss sampling of the area of carcinoma with FNA in thyroid nodules from clinically or radiological diagnosed multinodular goiter, especially, when there is no clinical or radiologic suspicion of malignancy. This rate is close to the 3.1%, 3.7%, 5.6%, 6% and 8% previously reported [11–15]. The risk of malignancy was originally reported for this category in the initial BSRTC definition at 0%–3% [16, 17].

We have also shown 33 (20.5%) of the FNA cytology being reported as AUS/FLUS (Bethesda category III). Of these 18 (54.5%) were reported malignant on histopathological examination and were either microcarcinoma or follicular variant PTC, or follicular carcinoma or classic PTC pT1b (>1cm<2cm) (Table 6). Interestingly, the reported malignancy rates for this category vary widely, from 50% [11, 12] to 69% [13] and 79% [18], while rates were much lower (10–30%) in the BSRTC definition [16, 17]. The lower rate of malignancy in category III according to the BSRTC definition report was probably due to the high number of cases included in their study as compared with the current study.

Nine (5.5%) cases of FNA cytology were reported as follicular neoplasm/suspicious for follicular neoplasm (Bethesda category IV). Of those, 6 (67%) were reported malignant on histopathological examination and were either microcarcinoma or follicular variant PTC (Table 6). This high rate was also reported by Lee et al. study 79% [18]. A lower rate was also seen in Mufti et al. 20% [11], and 52% in Zarin et al. study [12]. A lower rate (25–40%) has also been reported according to the BSRTC definition [16, 17]. The lower rate of malignancy in category IV in the BSRTC definition report is probably due to the high number of cases included in their study when compared with the current study.

The two indeterminate categories of III & IV are reported in the BSRTC definition report in about 15–30% and 25–40% of nodules respectively [16, 17]. They pose clinical challenges

**Table 6. Correlation between Bethesda category III and IV and histologic types of carcinoma.**

| Bethesda Category | Follicular Carcinoma | Follicular variant PTC | Microcarcinoma Classic Papillary carcinoma | Classic PTC |
|---|---|---|---|---|
| | | | pT1a (>1cm<2cm) | pT1b (>1cm<2cm) |
| III | 2 | 4 | 7 | 5 |
| IV | 0 | 2 | 3 | 1 |
| Total | 3 | 6 | 10 | 7 |

for endocrine physicians [19–21]. While the majority will have benign nodules, the malignancy risk is not trivial and typically triggers for additional investigations or treatment [20, 22]. Current strategy for such patients includes observation, repeat FNA with or without molecular markers, core needle biopsy or surgery (lobectomy or total thyroidectomy). As clear guidance on the best strategy is lacking, it largely depends on patient and or physician preference and experience, availability of molecular markers, cost, and the surgical expertise. Findings from US can also help in that nodules with suspicious lesions are typically managed surgically while those lacking features of malignancy might be observed periodically with serial US assessments [22]. In the current study, a high rate of risk of malignancy in these two categories has been identified, suggesting the requirement for a strict follow up of patients diagnosed as Bethesda III and IV with ultrasound guided FNA cytologic examination. The high rate of malignancy in Bethesda category III and IV seen in the current study could be related to the fact that many cases are found to be either microcarcinoma, that can easily be miss sampled, or follicular variant PTC, or follicular carcinoma (Table 6), which can easily be under diagnosed on FNA cytological examination either because of the lack of nuclear features of PTC or due to the difficulty in discriminating benign from malignant follicular cells proliferation as both may show increased cellularity and mild nuclear pleomorphism. In addition, in many follicular neoplasms, capsular invasion is needed to make the diagnosis of malignancy, a feature that is impossible to see in FNA cytological preparation.

We believe that FNA cytology is rather a screening tool more than a definite diagnostic tool and definite diagnosis can only be made by tissue biopsy. So it is very helpful in categorizing thyroid nodules according to their probable risk of malignancy from benign (least risk of malignancy) at one end to malignant at the other end of the spectrum. This makes the TBSRTC approach very appropriate in cytological categorization, management and follow-up of thyroid nodules. Each TBSRTC category has a diagnostic criteria and cytopathologists are keen to follow these criteria. Although, cytopathologists would prefer false negative rather than false positive diagnosis in thyroid FNA of difficult and ambiguous smears, but they are limited by the Bethesda cytopathological criteria of each category, sample adequacy, sample representability and sample processing. The way of smearing FNA samples on glass slides requires expertise and inadequate spread of lesional cells can affect the appearance of the cells and may affect the diagnosis. The cytopathological diagnostic criteria depends entirely on the nuclear features and if the nuclei are not visualized clearly in the smear, it will be difficult to come up with an appropriate diagnosis.

Twenty-four (14.6%) cases of FNA cytology were reported as suspicious for malignancy (Bethesda category V). Of those 23 were reported malignant by histopathological examination giving a 96% risk of malignancy. This is similar to what has been reported by Zarif et al. (95.6%) [12], and higher than was reported by the BSRTC report [16, 17]. The higher rate of Bethesda category V reported in the current study may be related to sampling issues and variable experience of aspiration techniques as well as differences in interpretation of findings. However, these differences should have no clinical consequences because the BSRTC recommendation for management of patients in category V and category VI is surgery.

Interestingly, 27 (16.5%) cases of FNA cytology were reported as malignant and all of them were proved to be malignant on histopathological examination making a 100% risk of malignancy of category VI, the same result reported by Mufti et al. [11], and similar to the risk of malignancy in the BSRTC report (97%–99%) [16, 17].

Thyroid US has been commonly used to identify the risk of malignancy in thyroid nodules, and help in making decisions on whether FNA is indicated or not. Previous studies have reported that several US features are associated with thyroid cancer and the majority are related to PTC [3].

We have shown 40 (24.8%) thyroid nodules have been reported as benign. Of these 2 were found malignant on histologic examination, giving a 5% risk of malignancy. This rate is significantly higher than the <1% previously reported [22–26]. This discrepancy is possibly due to the differences in samples and presence of microcarcinoma. We have also shown 21 (13%) of nodules reported are in the very low suspicious category by US. Of these 11 (52%) were reported malignant on histologic examination. This rate is significantly higher than the <3% previously reported [22–26]. The higher rate of malignancy in this category may be due to the difficulty of discriminating between benign and malignant US features.

Forty-one (25.6%) nodules were reported as low suspicious by US. Of these 15 were reported as malignant on histologic examination equating to a 36% risk of malignancy. This rate is significantly higher than the 5–10% previously reported [22–26]. The high rate of malignancy in this category may be due to the difficulty of discriminating between benign and malignant US features.

Fourteen (8.7%) nodules were reported as intermediate suspicious by US. Of those 14 were reported malignant on histologic examination equating to a 100% risk of malignancy. This rate is significantly higher than the 10–20% previously reported. [22–26]. The higher risk rate of malignancy of this US category in the current study is possibly related to a high threshold of suspicion by the radiologist in interpreting US images of these thyroid nodules.

Forty-five (27.9%) nodules were reported as highly suspicious by US. Of those 39 were reported malignant on histologic examination equating to an 87% risk of malignancy. This rate is within the range of previously reported risk of malignancy in this category 70–90% [22–26].

The comparison in the risk of malignancy shows significant differences between TBRSTC category VI and ATA US highly suspicious category. There is no evidence of false positive or false negative diagnosis in all FNA cytology reports of TBRSTC category VI (malignant) and the accuracy was 100%, while the accuracy of ATA US highly suspicious category was 87%. This suggests that FNA cytology is more accurate than US in predicting malignant thyroid carcinomas in the present study.

The limitations of this study include sample size as well as being a retrospective study. However, the studied material included all cases with thyroid nodules that fulfill the inclusion criteria in our tertiary care hospital during the period 2011–2019 and there is limited regional data comparing the risk of malignancy in thyroid nodules between FNA cytology and US examination [11, 27]. We have excluded TBRSTC category I from our study because they are non-diagnostic and the patients did not undergo thyroid surgery.

## Conclusions

Thyroid FNA cytological examination and US are key tools in predicting malignancy in thyroid nodules. Thyroid nodules with the diagnosis of Bethesda category III & IV run a high risk of malignancy thus more vigilance is required.

## Acknowledgments

We would also like to thank Dr. Klaus Van Gorkom, Department of Radiology, College of Medicine & Health Sciences, UAE University for his support in this project. We would also like to thank the Laboratory Medicine Department, Anatomic Pathology Division at Tawam Hospital, Al Ain City, UAE, for their support in this project. We would like thank Prof Chris Howarth for English editing.

## Author Contributions

**Conceptualization:** Suhail Al-Salam, Khaled M. Aldahmani, Juma Al Kaabi.

**Data curation:** Alia Al Dhaheri, Hayat Yahya, Duha Al Naqbi, Esraa Al Zuraiqi, Baraa Kamal Mohamed, Shamsa Ahmed Almansoori, Meera Al Zaabi, Aysha Al Derei, Amal Al Shamsi.

**Formal analysis:** Suhail Al-Salam, Charu Sharma.

**Investigation:** Suhail Al-Salam, Charu Sharma, Maysam T. Abu Sa'a.

**Methodology:** Suhail Al-Salam, Charu Sharma, Khaled M. Aldahmani.

**Project administration:** Juma Al Kaabi.

**Software:** Charu Sharma.

**Supervision:** Suhail Al-Salam, Juma Al Kaabi.

**Visualization:** Bachar Afandi, Khaled M. Aldahmani.

**Writing – original draft:** Suhail Al-Salam.

**Writing – review & editing:** Suhail Al-Salam, Charu Sharma, Bachar Afandi, Juma Al Kaabi.

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
