## [Decision Letter · Decision Letter 0]

1 Jan 2021

PONE-D-20-23164

Ultrasound-guided fine needle aspiration cytology and ultrasound examination of thyroid nodules in the UAE: a comparison

PLOS ONE

Dear Dr. Kaabi,

Thank you for submitting your manuscript to PLOS ONE. After careful consideration, we feel that it has merit but does not fully meet PLOS ONE’s publication criteria as it currently stands. Therefore, we invite you to submit a revised version of the manuscript that addresses the points raised during the review process.

We look forward to receiving your revised manuscript.

Kind regards,

Peter Dziegielewski, MD, FRCSC

Academic Editor

PLOS ONE

Journal Requirements:

2. Thank you for stating the following at the end of your manuscript:

'Funding

This work was supported by research grant (SURE Grant 31M352 and 31 M355), from Research Office, UAE University. The funders had no roles in study design, data collection and analysis, decision to publish, or prepare of the manuscript.'

'The funders had no role in study design, data collection and analysis, decision to publish, or preparation of the manuscript.'

Reviewers' comments:

Reviewer's Responses to Questions

**Comments to the Author**

1. Is the manuscript technically sound, and do the data support the conclusions?

Reviewer #1: No

Reviewer #2: Partly

2. Has the statistical analysis been performed appropriately and rigorously? 

Reviewer #1: I Don't Know

Reviewer #2: No

3. Have the authors made all data underlying the findings in their manuscript fully available?

Reviewer #1: Yes

Reviewer #2: Yes

4. Is the manuscript presented in an intelligible fashion and written in standard English?

Reviewer #1: No

Reviewer #2: Yes

5. Review Comments to the Author

Reviewer #1: The manuscript entitled "Ultrasound-guided fine needle aspiration cytology and ultrasound examination of thyroid nodules in the UAE: a comparison" reports a very interesting and important clinical problem. However, the relationship between US pattern, FNA diagnosis and histopathological structure of thyroid nodules is a subject of intensive research in many laboratories, and the issue investigated in this research is known from dozens of similar studies.

This research has serious flaws.

Firstly, the number of cases was small (164 patients treated between 2011-2019), and further studies with larger sample sizes are needed. What is particularly odd, the number of patients equals the number of lesions/nodules examined. Did all of the patients only have one nodule?

Secondly, there was selection bias, because patients included in this study underwent FNA and surgery, indicating that patients were not representative of the whole population.

Thirdly, this was a single-center, retrospective study, which may have reduced the statistical significance.

For instance, the reason of malignancy rate, especially in III and IV categories, being higher than that reported in TBSRTC, may be that cytopathologists from Tawam Hospital did not properly apply the TBSRTC classification criteria. Similarly, the high rate of malignances in III and IV categories, has been reported by Lee (Lee K, Jung CK, Lee KY, Bae JS, Lim DJ, et al. Application of Bethesda system for reporting thyroid aspiration cytology. Korean Journal of Pathology 2010; 44:521–7) and Park (Park JH, Yoon SO, Son EJ, Kim HM, Nahm JH, Hong S, et al. Incidence and malignancy rates of diagnoses in the Bethesda system for reporting thyroid aspiration cytology: An institutional experience. Korean Journal of Pathology 2014; 48:133–90). However, the explanation of this phenomenon given by Lee et al. is the pathologists’ mistake during classification to these categories. Different explanation was provided by Park. From the perspective of these authors there are two possible reasons for differences between malignancy rate in Korean study and official TBSRTC rates: “First, although the BSRTC guidelines recommend that patients with categories I or III diagnoses have a repeat FNA, in Korea, patients who have thyroid nodules that are strongly suspicious for malignancy in a clinical aspect undergo surgery without a repeat FNA, but a frozen section examination may be performed. Second, Korean patients tend to be more concerned about false positive results than false negative results, which may pressure cytopathologists to underdiagnose FNA cases to avoid making false positive diagnoses”.

Another surprising finding is the number of misdiagnosed cases of classic papillary carcinoma in III and IV categories in cytological examination – 16 out of 24 carcinomas in the group of 42 patients. This indicates that prior to possible repeated publication, a cytological evaluation of these smears, done by highly experienced specialists in thyroid cytopathology, should be conducted.

Minor Essential Revisions

Errors in abundances of groups in tables, and between tables 5 and 6.

Reviewer #2: This study was done in a single hospital in UAE, yielding 161 patients who were operated for thyroid nodules after undergoing both ultrasound and ultrasound guided fine needle aspiration cytology in a period of 8 years (from 2011-2019). It then proceeded to compare the results of each pre operative diagnostic examination with the final histopathologic diagnosis. Not unexpectedly, ultrasound-guided FNA cytology had a higher accuracy rate than ultrasound alone in identifying malignancies. They then proceeded to compare their findings with those of studies in Saudi Arabia and Turkey and found similar results with respect to higher BSRTC or ATA categories turning out malignant. Of concern was their finding that nodules found benign or low suspicious by ultrasound turned out malignant in 5% and 36%, respectively.

The main limitation of this study is that the sample is restricted to the retrospective review of surgical patients only. Thus it is impossible to evaluate the true accuracy of both diagnostic assessments and to apply its results to the majority of non-operative patients with clinically apparent thyroid nodules. Ideally, sensitivity, specificity and accuracy are determined by applying both the test being evaluated and the reference standard on a group of patients regardless of the test result. In this study, the decision to operate on these patients with nodules would have probably been based on the results of either one or both tests, thus biasing the study.

Because of the inherent limitation of their study design the authors must be careful not to apply analytical tools which are more commonly used for diagnostic validation studies. Their conclusions must likewise be tempered by this limitation.

6. PLOS authors have the option to publish the peer review history of their article (what does this mean?). If published, this will include your full peer review and any attached files.

Reviewer #1: No

Reviewer #2: **Yes: **Jose M. Acuin

---

## [Author Response · Author response to Decision Letter 0]

3 Feb 2021

Response to Reviewers’ comments

Reviewer 1

Many thanks for your valuable comments. Your comments are herein addressed point by point. We agree with you that there are published papers addressing similar issue, but nothing has been published from this part of the world. In this work, we would like to share our experience in FNA cytology and US of thyroid nodules with the international scientific community. 

1. The number of cases was small (164 patients treated between 2011-2019), and further studies with larger sample sizes are needed. 

Response to reviewer comment

We agree with you that sample size was small. This point has been mentioned in the discussion as a limitation of this study. These are all the cases that were diagnosed by thyroid FNA cytology and US and were followed by surgery during the period 2011-2019. Tawam hospital is a tertiary care center and the main oncology center in the UAE; hence, most of the cases were referred to this hospital from different parts of the country.

2. What is particularly odd, the number of patients equals the number of lesions/nodules examined. Did all of the patients only have one nodule?

Response to reviewer comment

Not all the patients have one nodule. Based on the clinical and US findings, the FNA was done on the most suspicious thyroid nodule, which was followed by histopathological study of the resected specimens. That is why the number of cases have same number of examined nodules by FNA and surgical resection. 

3. There was selection bias, because patients included in this study underwent FNA and surgery, indicating that patients were not representative of the whole population.

Response to reviewer comment

We need histopathology to see the risk of malignancy in each Bethesda category as well as in American thyroid association ultrasonographic categories besides measuring the accuracy of each procedure. So, we have studied all the cases that had thyroidectomy following US thyroid and FNA during the period 2011-2019. We think that there was no selection bias in our selection criteria because we include all cases in our center that fulfill the inclusion criteria. Tawam hospital is a tertiary care hospital and the main oncology hospital in the UAE; hence, most of the cases were referred to this hospital from different parts of the country. 

4. This was a single-center, retrospective study, which may have reduced the statistical significance.

Response to reviewer comment

We agree with you that this is a retrospective study from one center. Tawam hospital is a tertiary care center and the main oncology center in the UAE; hence, most of the cases were referred to this hospital from different parts of the country. Besides, we would like to share our experience in the diagnosis and management of thyroid nodules with the scientific community. 

5. The reason of malignancy rate, especially in III and IV categories, being higher than that reported in TBSRTC, may be that cytopathologists from Tawam Hospital did not properly apply the TBSRTC classification criteria. Similarly, the high rate of malignances in III and IV categories, has been reported by Lee (Lee K, Jung CK, Lee KY, Bae JS, Lim DJ, et al. Application of Bethesda system for reporting thyroid aspiration cytology. Korean Journal of Pathology 2010; 44:521–7) and Park (Park JH, Yoon SO, Son EJ, Kim HM, Nahm JH, Hong S, et al. Incidence and malignancy rates of diagnoses in the Bethesda system for reporting thyroid aspiration cytology: An institutional experience. Korean Journal of Pathology 2014; 48:133–9). However, the explanation of this phenomenon given by Lee et al. is the pathologists’ mistake during classification to these categories. Different explanation was provided by Park. From the perspective of these authors there are two possible reasons for differences between malignancy rate in Korean study and official TBSRTC rates: “First, although the BSRTC guidelines recommend that patients with categories I or III diagnoses have a repeat FNA, in Korea, patients who have thyroid nodules that are strongly suspicious for malignancy in a clinical aspect undergo surgery without a repeat FNA, but a frozen section examination may be performed. Second, Korean patients tend to be more concerned about false positive results than false negative results, which may pressure cytopathologists to underdiagnose FNA cases to avoid making false positive diagnoses”.

Another surprising finding is the number of misdiagnosed cases of classic papillary carcinoma in III and IV categories in cytological examination – 16 out of 24 carcinomas in the group of 42 patients. This indicates that prior to possible repeated publication, a cytological evaluation of these smears, done by highly experienced specialists in thyroid cytopathology, should be conducted. 

Response to reviewer comment

We thank you for providing the two references (Lee K et al. and Park JH ) whom have high rates of TBSRTC category III and IV in their studies were similar to ours. We have referred to both studies in the discussion of our manuscript as (reference 17 and 12 respectively). 

There are many explanations for the high rate of malignancy TBSRTC category III and IV and we have stated that in the discussion section of our manuscript “The high rate of malignancy in Bethesda category III and IV seen in the current study could be related to the fact that many cases are found to be either microcarcinomas, that can easily be miss-sampled, or follicular variant PTC, or follicular carcinoma (Table 6), which can easily be under diagnosed on FNA cytological examination either because of the lack of nuclear features of PTC or due to the difficulty in discriminating benign from malignant follicular cells proliferation as both may show increased cellularity and mild nuclear pleomorphism. In addition, in many follicular neoplasms, capsular invasion is needed to make the diagnosis of malignancy, a feature that is impossible to see in FNA cytological preparation.

We believe that FNA cytology is rather a screening tool more than a definite diagnostic tool and definite diagnosis can only be made by tissue biopsy. So it is very helpful in categorizing thyroid nodules according to their probable risk of malignancy in a spectrum wise from benign (least risk of malignancy) at one end to malignant at the other end of the spectrum. This makes TBSRTC system very appropriate in cytological categorization, management and follow-up of thyroid nodules. Each TBSRTC category has a diagnostic criteria and cytopathologists are keen to follow these criteria. Although, cytopathologists would prefer false negative more than false positive diagnosis in thyroid FNA of difficult and ambiguous smears, but constantly they are limited by the Bethesda cytopathological criteria of each category, sample adequacy, sample representability and sample processing. The way of smearing FNA samples on glass slides requires certain expertise and inadequate spread of lesional cells can affect the appearance of the cells and may affect the diagnosis. The cytopathological diagnostic criteria depends entirely on the nuclear features and if the nuclei are not visualized clearly in the smear, it will be difficult to have appropriate diagnosis.

Cytopathologists training and experience are also very important limiting factors, and we can assure you that our cytopathologists at Tawam hospital are expert in this field.

This paragraph is added to the discussion.

Minor Essential Revisions

Errors in abundances of groups in tables, and between tables 5 and 6.

Response to reviewer comment

In figure 5 correlation between Bethesda categories and risk of malignancy, all the numbers in the table represent the percentage and are not the number of cases.

While, the number in figure 6 represents the actual number of malignant histology diagnosis in resection specimens of TBSRTC category III and IV cases.

Reviewer #2: 

The main limitation of this study is that the sample is restricted to the retrospective review of surgical patients only. Thus, it is impossible to evaluate the true accuracy of both diagnostic assessments and to apply its results to the majority of non-operative patients with clinically apparent thyroid nodules. Ideally, sensitivity, specificity and accuracy are determined by applying both the test being evaluated and the reference standard on a group of patients regardless of the test result. In this study, the decision to operate on these patients with nodules would have probably been based on the results of either one or both tests, thus biasing the study. Because of the inherent limitation of their study design the authors must be careful not to apply analytical tools which are more commonly used for diagnostic validation studies. Their conclusions must likewise be tempered by this limitation.

Response to reviewer comment

Many thanks for your valuable comments. We totally agree with your comments. We have mentioned these limitations in the discussion. In addition, our conclusions will be tempered by this limitation

---

## [Editor Report · Decision Letter 1]

15 Feb 2021

Ultrasound-guided fine needle aspiration cytology and ultrasound examination of thyroid nodules in the UAE: a comparison

PONE-D-20-23164R1

Dear Dr. Kaabi,

We’re pleased to inform you that your manuscript has been judged scientifically suitable for publication and will be formally accepted for publication once it meets all outstanding technical requirements.

Kind regards,

Peter Dziegielewski, MD, FRCSC

Academic Editor

PLOS ONE

Additional Editor Comments (optional):

Thank you for addressing the reviewers comments and critiques.
---

## [Editor Report · Acceptance letter]

29 Mar 2021

PONE-D-20-23164R1 

Ultrasound-guided fine needle aspiration cytology and ultrasound examination of thyroid nodules in the UAE: a comparison 

Dear Dr. Kaabi:

I'm pleased to inform you that your manuscript has been deemed suitable for publication in PLOS ONE. Congratulations! Your manuscript is now with our production department. 

Kind regards, 

on behalf of

Dr. Peter Dziegielewski 

Academic Editor

PLOS ONE